# A Review of Deep Reinforcement Learning Approaches for Smart Manufacturing in Industry 4.0 and 5.0 Framework

**Alejandro del Real Torres** [1], **Doru Stefan Andreiana** [2,*], **Álvaro Ojeda Roldán** [2], **Alfonso Hernández Bustos** [2] **and Luis Enrique Acevedo Galicia** [2]

1   School of Engineering (ETSI), Systems and Automation Department, University of Sevilla, 41092 Sevilla, Spain
2   IT Department, IDENER, 41300 Sevilla, Spain
*   Correspondence: doru.stefan@idener.es

**Abstract:** In this review, the industry's current issues regarding intelligent manufacture are presented. This work presents the status and the potential for the I4.0 and I5.0's revolutionary technologies. AI and, in particular, the DRL algorithms, which are a perfect response to the unpredictability and volatility of modern demand, are studied in detail. Through the introduction of RL concepts and the development of those with ANNs towards DRL, the potential and variety of these kinds of algorithms are highlighted. Moreover, because these algorithms are data based, their modification to meet the requirements of industry operations is also included. In addition, this review covers the inclusion of new concepts, such as digital twins, in response to an absent environment model and how it can improve the performance and application of DRL algorithms even more. This work highlights that DRL applicability is demonstrated across all manufacturing industry operations, outperforming conventional methodologies and, most notably, enhancing the manufacturing process's resilience and adaptability. It is stated that there is still considerable work to be carried out in both academia and industry to fully leverage the promise of these disruptive tools, begin their deployment in industry, and take a step closer to the I5.0 industrial revolution.

**Keywords:** deep reinforcement learning; smart manufacturing; industry 4.0; industry 5.0; sim-to-real transfer; path planning; scheduling; process control; robotics; maintenance; energy management

## 1. Introduction

Roughly a decade ago, industry 4.0 (I4.0) emerged as the term to define the fourth industrial revolution. Its objective is the transition from the mass production automation of the third industrial revolution to more efficient and flexible production [1]. It can be defined as a technology-driven revolution focusing on further automation and the digitalisation of industrial processes. This results in smart factories, which make use of improved technologies, such as artificial intelligence (AI), Internet of Things (IoT), cloud computing and cyber-physical systems (CPS) [2]. However, it lacks a human-centric and sustainability-centred vision. Moreover, the COVID-19 crisis revealed some deficiencies in global industrial production, which lacks enough flexibility to deal with abrupt changes in production demand [3]. For this reason, the term industry 5.0 (I5.0) has been introduced [4]. This new concept strengthens and complements the objectives of I4.0 through a human-centric, sustainable and resilient industry, reinforcing the contribution of industry to worker welfare and green transition. To this end, it combines the advances in I4.0 technologies in terms of digital twins, CPS, Big Data and AI, among others, with innovative technologies that have surged in the last years [5]. all in all, with a human and sustainable-centred perspective [6].

In 2021, manufacturing recovered to pre-pandemic levels of activity, generating approximately 17% of the gross domestic product (GDP) on average around the world [7] and 14.9% in the European Union, making it the most important industrial activity at the economic level [8].

Independently of the sector, manufacturing comprises many processes, from planning and scheduling to executing physical operations in the production line until the product is ready for distribution [9]. Among these processes, there are tasks involving production scheduling, assembly, decision support systems and path planning. Currently, many of them are carried out by digital systems and robots thanks to the automation of factories, improving their efficiency [10]. However, applying artificial intelligence, in particular machine learning (ML), takes a step forward in this enhancement. Without being explicitly programmed, machine learning algorithms endow automatons with cognitive capabilities that allow them to learn a task [11]. However, the bulk of these algorithms require data in order to learn, and it is not always possible to obtain accurate data in some industrial settings. Reinforcement learning (RL) is a machine learning paradigm that is ideal since its algorithms immediately learn from interaction with the environment. Additionally, the use of deep neural networks (DNNs) with RL algorithms gave rise to deep reinforcement learning (DRL), whose algorithms are capable of learning more complex tasks [12]. Furthermore, those algorithms are relevant for both I4.0 and the upcoming I5.0 since they align with the objective of industry 5.0 easily adapting to a more human-centred approach [13,14].

As reflected in most of the reviews concerning smart manufacturing, I4.0 and I5.0, AI is identified as a key enabling technology. However, AI is a huge study field, and the majority of those reviews do not go deep into how and what to implement given a specific problem. Furthermore, the results in that direction are even scarcer when focusing on more specific AI fields such as DRL. In this sense, the present paper provides a review of the most commonly used DRL algorithms in manufacturing processes, including their main characteristics and performance, real applications and implementation. Therefore, the paper is intended to serve as a guideline for the development and improvement of factories in line with industry 4.0 and 5.0, promoting the use of DRL techniques and algorithms.

The structure of the paper is as follows. Firstly, the fundamentals of reinforcement learning are briefly explained. Section 2 depicts the search performed for publications on this topic. Section 3 outlines current DRL algorithms and their classification according to their features. Section 4 illustrates the use of DRL algorithms in manufacturing. Section 5 demonstrates the current training techniques for those algorithms and its implementation in real-world tasks. Finally, conclusions and trendy lines of research are exposed.

*Reinforcement Learning*

In the early history of reinforcement learning, there were three threads; the first one focused on learning by trial and error; the second one centred on the problem of optimal control; and the third one surged later on, based on ideas from the first two, concerned temporal–difference methods. All of them came together in the late 1980s to give birth to the modern field of reinforcement learning [15]. Nowadays, reinforcement learning has been consolidated as one of the three main machine learning paradigms, together with supervised and unsupervised learning [16,17].

Reinforcement learning algorithms are based on an iterative learning process. The learning process is based on trial and error and the interaction between an agent and an environment [18]. This interaction is modelled as a Markov Decision Process (MDP), a concept first introduced by Bellman R. E. in 1957 [19]. Through this idea, the interaction is reduced to three signals: state $s$ (the current situation of the environment); action $a$ (operation or decision taken by the agent based on the state and its experience); and reward $r$ (numerical feedback that the environment returns to the agent to indicate how good or bad is the action taken by the agent) [20]. Figure 1 illustrates this interaction.

The objective of the agent is to maximise the accumulative reward in the long run, which entails learning the task [21]. This is reflected within the policy of the agent, $\pi$, which determines which action is more suitable to take in each state. This policy is updated and improved as the agent interacts with the environment and gains experience [22]. The acquired knowledge is represented in a value function, which can be defined in two ways:

- State-value function for policy $\pi$ ($V_\pi(s)$), which assigns the expected accumulative reward to each state as the agent will receive if the interaction starts in state $s$ and follows the policy $\pi$.
- Action-value function for policy $\pi$ ($Q_\pi(s,a)$)), which determines the expected accumulative reward for each action-state pair as the agent will receive if it starts the in state $s$, takes action $a$ and follows the policy $\pi$ thereafter.

In the learning process, the value function and the policy are updated and improved regarding each other. Under this, the exploration–exploitation problem exists, and a trade-off must be found [23,24]. On the one hand, exploration of new actions is necessary to learn alternative paths to achieve the goal and learn the task, whereas exploitation leverages the acquired knowledge to maximise the accumulative reward [25]. On the other hand, exploitation consists of applying the knowledge learned and mostly taking the optimum action known [26].

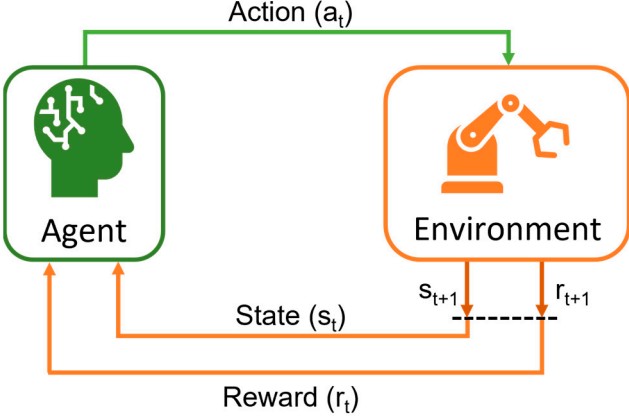

**Figure 1.** Structure of Markov Decision Process.

Firstly, RL algorithms go through an exploratory phase to learn the dynamics of the environment. In this sense, there are several exploration techniques. The most common technique is a random exploration which usually gives great results, as reflected through impressive performances in self-driving cars [27], autonomous landing [28], Atari games [29], Mujoco simulator [30], controller tuning [31] and much more. There are also complex techniques, such as reward shaping [32,33], where the algorithm designer arbitrarily modifies the agent's rewards. However, this technique highly depends on the designer's experience and knowledge of the problem. Errors in reward shaping may lead to infinite repetition of action [34] or no actions at all [35]. An extensive analysis concerning the exploratory techniques and their benefits and drawbacks can be found in Pawel L. et al.'s (2022) survey [36].

Secondly, once the agent explores the environment and learns the consequence of its actions, it passes to exploit that knowledge. However, depending on the problem handled, the agent usually maintains part of its exploratory behaviour just to ensure that the policy of actions followed is still the best. The balance between exploration and exploitation is still an open issue under investigation since there is no unique and perfect solution, but every problem has its own solution [37–39].

## 2. Paper Research and Evaluation

Given the plethora of DRL applications in manufacturing, a robust methodology that allows for the gathering and analysis of all of them becomes necessary in order to obtain a complete review of the current situation in this field. For the elaboration of this review, a cutting-edge method of bibliometric analysis has been performed. This method enables obtaining numerous statistics and connections between the publications, detecting the most relevant DRL applications in manufacturing together with the most active researchers. These publications address the current trends in this field, which deal

with crucial challenges and limitations in manufacturing. Thanks to this methodology, striking DRL techniques and algorithms are identified, which will lead to the improvement in the efficiency of industrial processes in the near future.

Bibliometric analysis is a type of research that aids in the understanding of trends in the scientific publications field [40]. This analysis enables extracting useful information, such as growth rates, co-occurrences, co-authorship, outputs per country and collaborations. Thanks to this information, it is possible to predict where the industry is heading and how.

To better aid in the comprehension and extraction of information from the bibliometric research, a visual analysis has been performed in the form of a citation network [41]. This tool provides a visual understanding of the connection between publications, as well as other metrics related to the references of each paper, such as most cited publications and the appearance of keywords in the papers [42]. The citation network complements the bibliometric analysis, achieving a more holistic approach to the research.

### 2.1. Methodology

- To perform the bibliometric analysis, the Bibliometrix tool was used [43], which allows for the quick extraction of bibliographic information from a given bibliographic export. Such bibliographic export is obtained from a search engine of the academic publisher https://www.scopus.com (accessed on the 15 November 2022). To this end, the main research question must be formulated to define the bibliometric analysis: *What are the main deep reinforcement learning approaches used in manufacturing processes?* Thus, this question must be translated in a search through this engine. This search must be carried out with keywords and logic operators that delimit the search field on which this review focuses. The most accurate search was performed on 1 November 2022 with the following query: *TITLE-ABS-KEY (("DEEP REINFORCEMENT LEARNING" OR DRL OR "DEEP RL") AND (MANUFACTURING OR ROBOTS OR "PRODUCTION SYSTEM" OR AUTOMATION) AND ("MODEL-FREE" OR "MODEL-BASED" OR "ON-POLICY" OR "OFF-POLICY")).*
- In this way, the result of the search will return documents that focus on the different approaches based on DRL algorithms and techniques in manufacturing processes, such as model-free, model-based, on-policy or off-policy algorithms. The list of scientific publications contains the necessary information to answer the previous research question.
- The next step involves creating the citation network, for which the bibliographic export is processed through a script programmed in R [44]. For its visualisation, an open-source software, Cytoscape [45] was employed. Furthermore, the methodology to remove unwanted information, based on Zuluaga et al.'s (2016) article [46] was implemented to achieve a cleaner visualisation of the results.

### 2.2. Bibliometric Analysis

From the previous query, a total of 244 documents were returned (of which one was excluded due to it not staying within the topic), ranging in publication date from 2016 to 2023. From the initial period up to 2021, an annual growth rate of 133.22% was found, making this research topic very important to the field, as illustrated by Figure 2. Furthermore, from the entire range of publication dates, it was found that there is an international co-authorship of 25.1%. In addition to measuring the co-authorship, this indicator includes hidden relationships between the co-authors of a scientific paper, such as motivation and the research rank of their institutions [47]. This level of international co-authorship involves a high degree of international collaboration and a positive contribution to the citation impact [48]. As a result, there is a noteworthy flow of researchers between institutions in different parts of the world, enabling deep investigation of DRL applications in manufacturing. This is illustrated in Figure 3, where the most active researchers in the field are linked to the most relevant topics they have investigated and the research institutions they belong to. It can be noted that most of the researchers have belonged to

several research institutions, showing the close collaboration between them, and they have participated in several scientific publications on different topics within this field. These collaborations allow progress in the development of new DRL technologies in the industry.

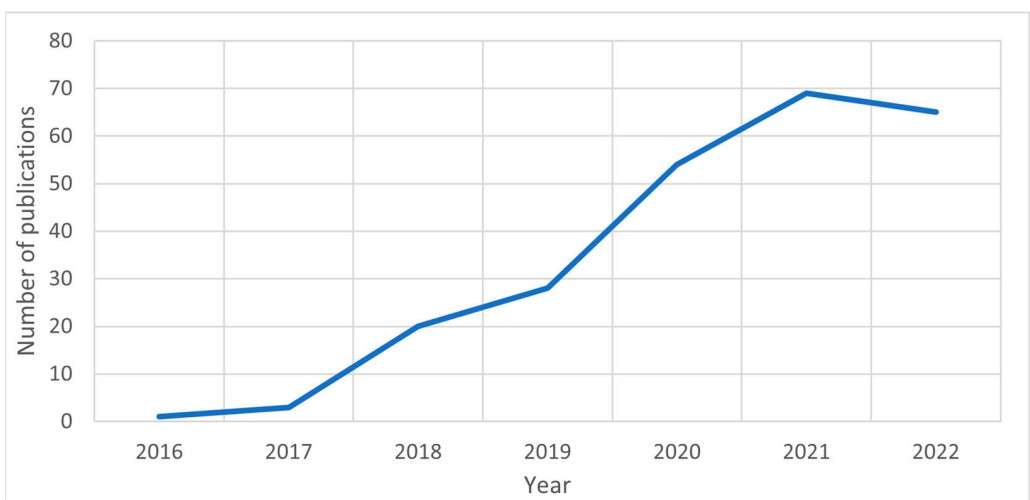

**Figure 2.** Annual scientific production of the query.

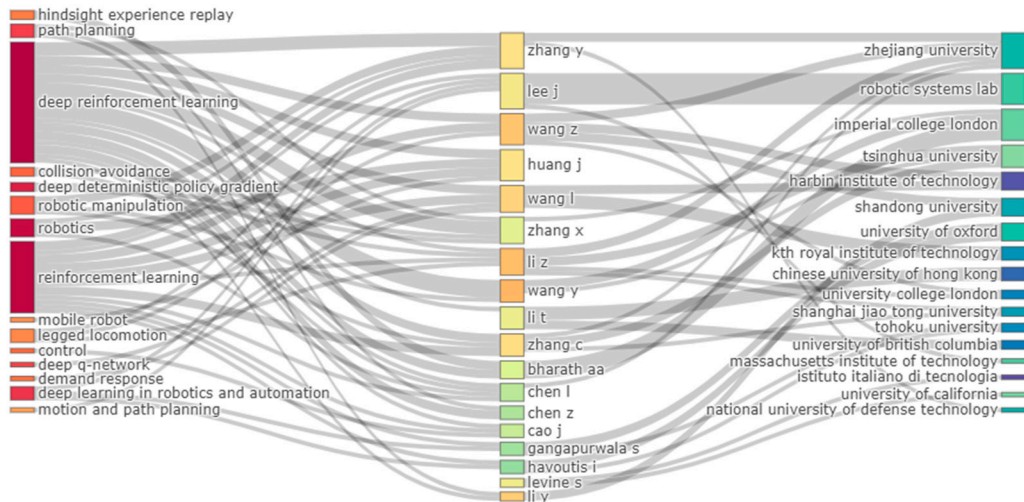

**Figure 3.** Three-field plot: author keywords on the left, authors in the middle and affiliations on the right.

The institutions shown in Figure 2 are leading in deep reinforcement learning research. If this is translated geographically, it can be observed that most of them belong, in this order, to China, the United States and the United Kingdom. Figure 4 depicts the collaborations between institutions by country, where the strongest research relationships in this field emerge. The strong and stable relationship between American and Chinese institutions is noteworthy, with a flow of researchers in both directions. In fact, for China, the United States is the only country with which close collaboration is observed. However, the United States does have collaborations with institutions in European countries, such as Imperial College London and Oxford University from the United Kingdom and the Robotic Systems Lab belonging to ETH Zürich (Switzerland). Lastly, although not many active researchers from Canadian institutions have been detected, the analysis shows a large number of collaborations between those countries.

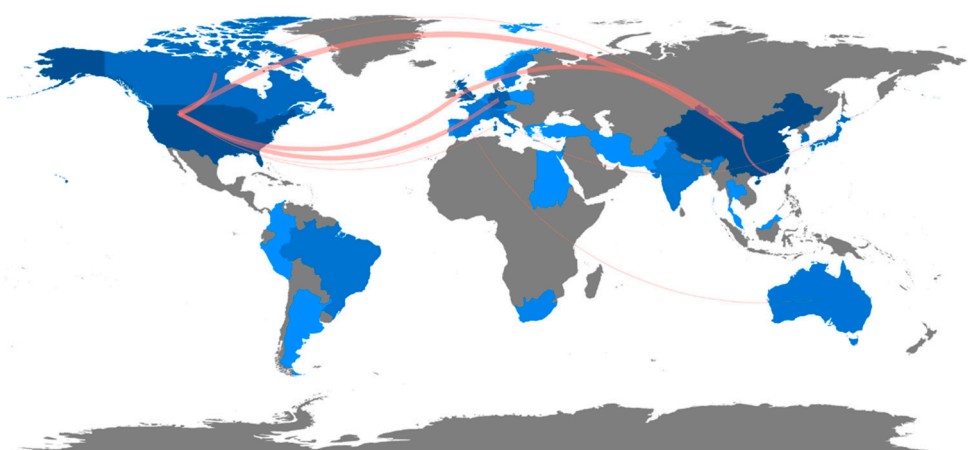

**Figure 4.** Collaboration world map.

Finally, an analysis of the journals was carried out following Bradford's Law, a law of diminishing returns in which journals are divided into three zones depending on the frequency of citation [49]. Figure 5 shows in descending order the essential sources of information when looking for DRL knowledge.

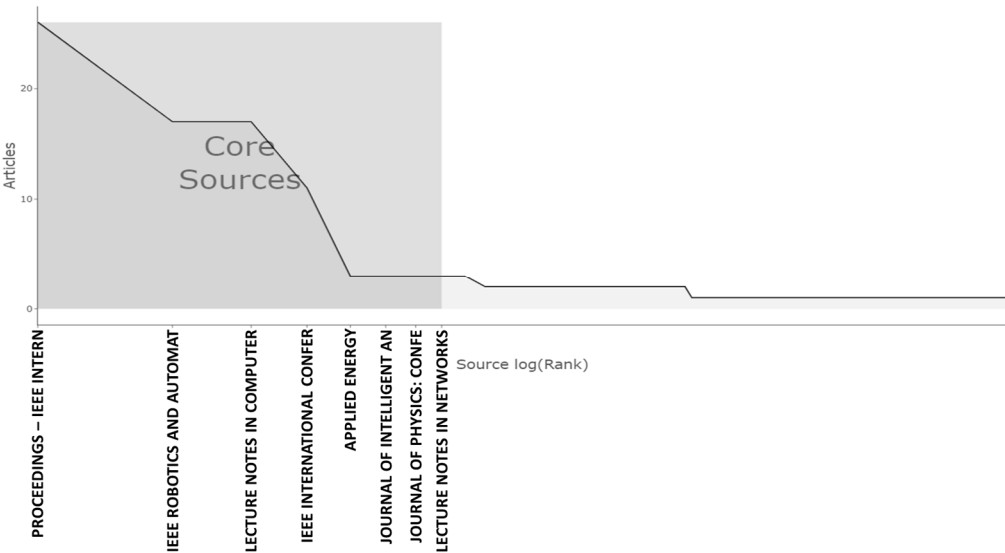

**Figure 5.** Source clustering through Bradford's Law.

The most cited document was published by Gu et al. (2016) [50], which presents a DRL algorithm applied to robot manipulation tasks. It was cited globally 701 times, making it the most prolific document in the search.

Regarding the appearance of the main approaches to DRL, of the 243 documents: 131 of them mention a model-free approach (53.91%), 113 model-based (46.50%), 30 off-policy (12.35%) and 14 on-policy (5.76%) approaches. Moreover, 35 documents mention both model-based and model-free approaches (14.40%); one mentions both model-based and on-policy approaches (0.41%); one mentions model-free and on-policy approaches; four mention model-free and off-policy approaches (1.65%); and four mention on-policy and off-policy approaches. Overall, it seems that the DRL applications in the industry are more focused on the need for a model of the environment, highlighting the vast presence of model-free and model-based approaches. Nonetheless, although any algorithm can be classified as on-policy and off-policy, this characteristic is not highlighted in the title, abstract or keywords of the scientific publications analysed. These data are illustrated in the form of a Venn diagram in Figure 6:

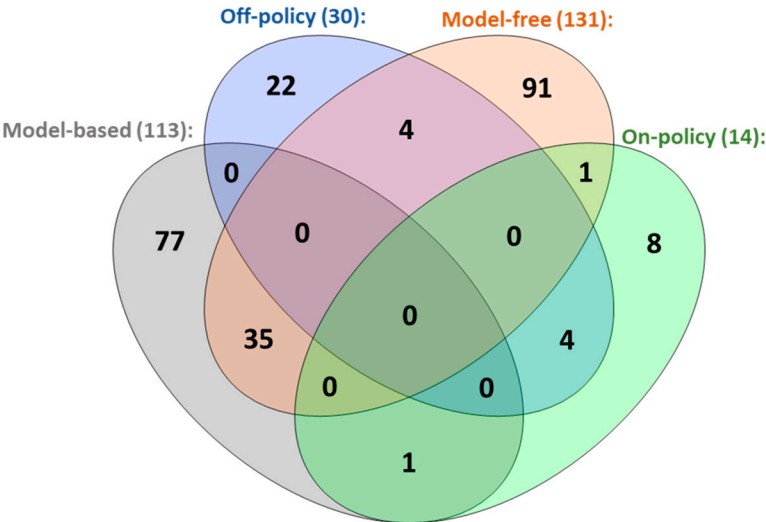

**Figure 6.** Venn diagram of DRL approaches occurrences in the research query.

### 2.3. Citation Network

To create the citation network, data from the references of each document in the search were used, making each publication a node in the network. Two nodes are linked with an edge if one of the corresponding documents cites the other one. The resulting citation network comprises 485 nodes (documents) and 863 edges (citations). It is shown in Figure 7, where colour indicates the year of publication of the document pertaining to that node, and size indicates the number of times it has been cited:

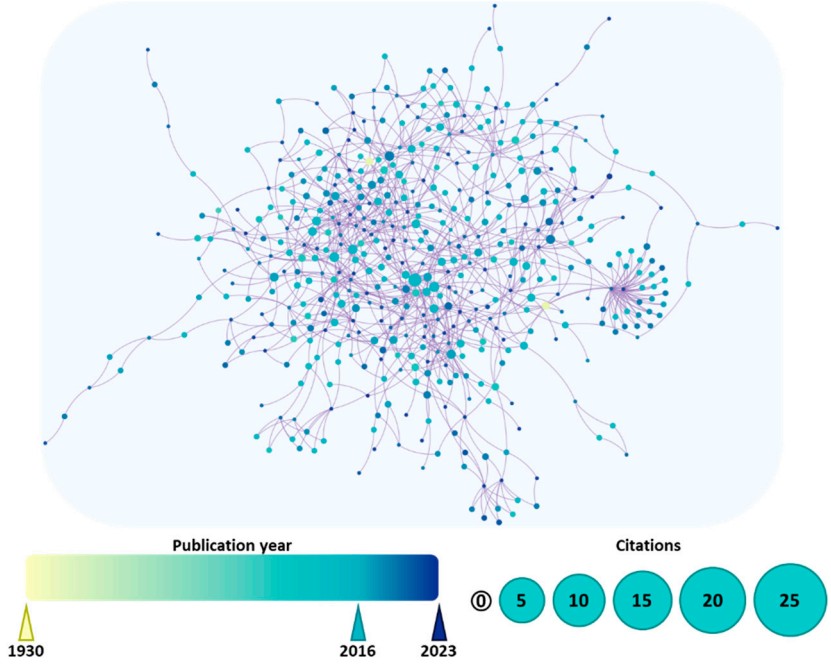

**Figure 7.** Citation network with colour and size legends.

The most cited document in the network was by Mnih et al. (2015) [29], cited 25 times locally (within the citation network) and 12,691 times globally (in the database search). They used a Deep Q-Network to demonstrate its capabilities against previous algorithms in beating Atari 2600 games. The second most cited document was by Levine et al. (2016) [51], cited locally 14 times and globally 1354 times. They developed a method to learn certain deep convolutional neural network policies to determine whether it is better to train the perception and control systems jointly end-to-end or train each component separately.

The oldest document in the network was by Uhlenbeck et al. (1930) [52], in which they discussed the Brownian motion. The second oldest document was by Maciejewski et al. (1985) [53], published in 1985, in which they discussed dynamic obstacle avoidance for manipulators with motion control and multiple goals.

To analyse the appearance of keywords in the citation network, the following search queries were performed and grouped accordingly: (i) *manufactur\* or automat\* or "production system"*, (ii) *robot\**, (iii) *\*polic\** and (iv) *"reinforcement learning"*. Note that the asterisks are used to replace multiple characters anywhere in a word. For example, the query "*\*polic\**" would return documents containing any of the following terms: policy, policies, on-policy and off-policy, among others.

The results can be seen in Figure 8, where size indicates the number of times the pertinent document has been cited and colours are chosen based on the grouping of the Venn diagram:

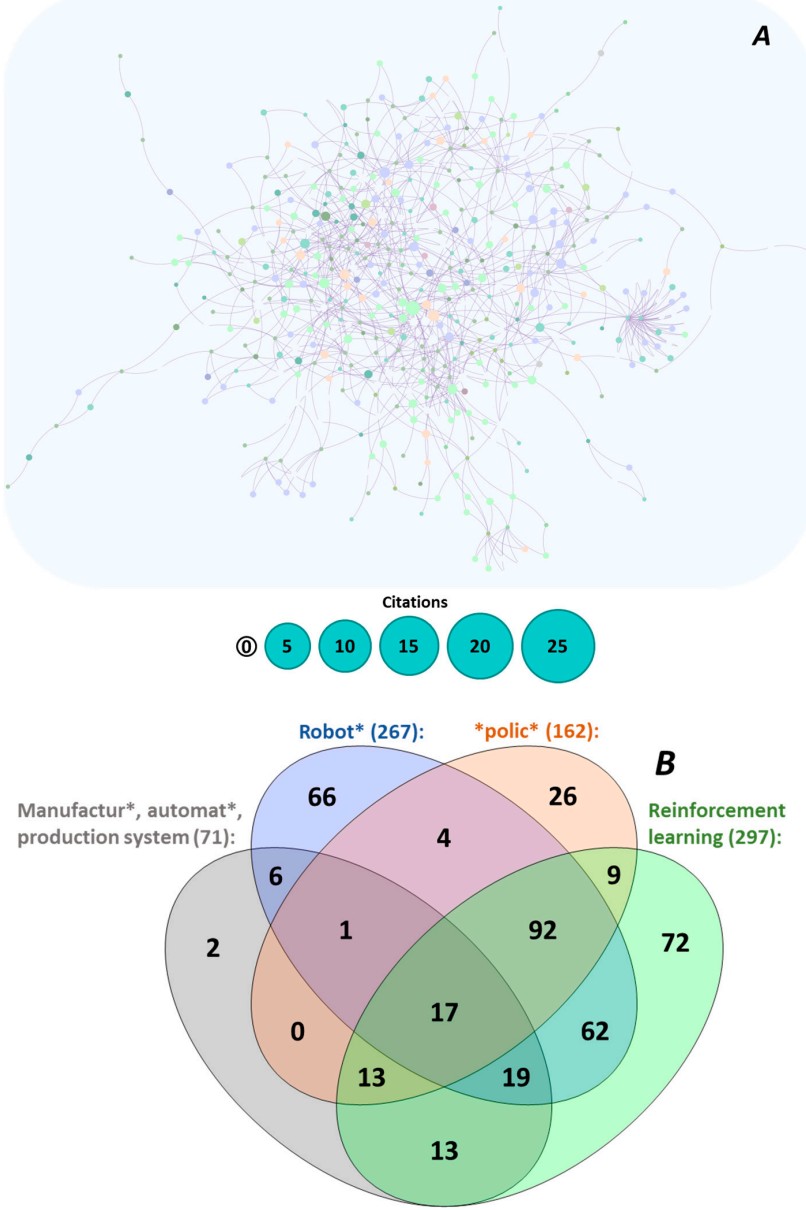

**Figure 8.** (**A**) Citation network coloured based on the appearance of selected words with size legend; (**B**) coloured Venn diagram of keywords. The asterisks are used to replace multiple characters anywhere in a word.

Of the 485 documents in the network, 402 satisfy any of the previous search queries. Overall, 71 documents (14.64%) belong to the first group, 267 (55.05%) to the second group, 162 (33.40%) to the third group and 297 (61.24%) to the fourth group. Only 17 documents (3.51%) are part of the four groups simultaneously. As can be seen, the documents from the original search query rely mainly on other documents that encompass robotics and reinforcement learning, meaning that the application of DRL in manufacturing processes is a relatively new topic that is being developed from the ground up with its base built upon previous works in robotics and reinforcement learning. This is also demonstrated by the fact that the original bibliographic export comprised mainly documents from 2016 onwards. Then, it is to be expected that in the following years, future work will be built upon today's works with exponential growth as new developments in DRL occur.

*2.4. Analysis of Results*

The bibliometric analysis reveals that deep reinforcement learning is being increasingly applied to manufacturing processes each year, thus making this a very prolific research field with many novel applications to I4.0 and I5.0, which are of international interest. This field sparks collaboration between institutions of different countries, making knowledge a global resource with which new and improved methods arise. As such, the necessity to keep researching, expanding and improving this field is of crucial importance.

Of the studied algorithms, it was found that both model-based and model-free approaches equally appear in scientific publications. Nonetheless, the on-policy/off-policy feature seems to be less relevant as it is less present in the analysis. Therefore, it can be deduced that the latter classification is important at a theoretical level, and the classification according to the availability of a model of the environment is more related to the applicability of the algorithm to the industry

Regarding the most important document of the research output for the query in terms of citations, it was an article by Gu et al. (2016) [50], demonstrating the use of an off-policy DRL method that enables the use of robotic manipulators in complex 3D environments, as well as making it possible to test on real physical robots. Understandably, this is a key document, as it discusses the application of DRL in complex and real systems.

As per the citation network, it was found that the most important document was by Mnih et al. (2015) [29] due to the fact that it was the most cited document locally and very prolific globally. As it compares the back-then novel Deep Q-Network (DQN) to previous algorithms, it is a critical document when arguing why a DQN is used compared to other methods, so it should be taken into account for future developments.

It has been seen from the oldest documents concerning Brownian motion and dynamic obstacle avoidance of manipulators, from which a base was erected, to advances in DQN and DRL applied to manipulators in complex 3D environments. All in all, as a crucial topic to research, DRL keeps growing at an accelerating pace in manufacturing environments internationally. As such, it is essential to learn about the inner workings of the main DRL algorithms and their place in the industry, a topic that will be discussed in the following sections.

## 3. Deep Reinforcement Learning

To address higher dimensional and more complex problems, deep neuronal networks (DNNs) were incorporated into RL, leading to deep RL (DRL) [54]. DNNs are used as function approximators to estimate the policy and value function. Moreover, leveraging their capacity to compact input data dimensionality, hence more complex observations, such as images and non-linear problems, can be processed [55–57]. This DRL field started with the Deep Q-Networks (DQN) algorithm [29] which has exponentially increased over the last few years. This section describes the catalogue of DRL algorithms, including their primary properties and classification schemes, illustrated in Figure 9.

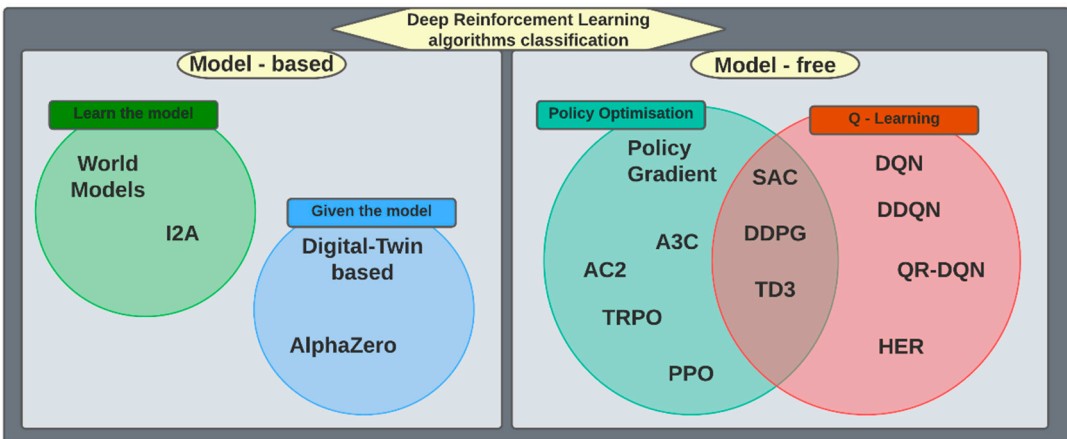

**Figure 9.** Classification of deep reinforcement learning algorithms.

Figure 9 depicts the most extended classification of DRL algorithms [58]. The main grouping is based on the available information about the dynamics of the environment, which determines the learning process of the agent.

On the one hand, model-based algorithms can be distinguished. These algorithms have access to information on the environment dynamics, including the reward function, which allows the agent to estimate how the environment will react to an action [59]. Typically, these algorithms are integrated with metaheuristics and optimisation techniques [60]. Moreover, they are particularly good at solving high-dimensional problems, as reflected in Aske P. et al. (2020) [61] survey. Furthermore, those methods reflect a higher sample efficiency, as reflected through empirical [62,63] and theoretical [64] studies. A complete overview concerning model-based DRL is presented by Luo, F. et al. (2022) [65] in their survey. Inside model-based DRL algorithms [59], there are two different situations depending on if the model is known or not.

Concerning the first group, if the model is known, this knowledge is used to improve the learning process, and the algorithm is integrated with metaheuristics, planning and optimisation techniques. However, since the environments usually have large action spaces, the application of these techniques is highly resource demanding. Thus, a complete optimisation of the learning process cannot be carried out. Moreover, although there are no algorithms defined as such, except for well-known algorithms such as Alpha Zero [66] and Single Agent [67], most of them are adapted to the application and the characteristics of the model environment. In recent years, DRL model-based algorithms that make use of digital twin models may be highlighted, such as the algorithms presented by Matulis and Harvey (2021) [68] and Xia et al. (2021) [69]. Therefore, in the implementation of the model-based algorithms, the following aspects must be addressed:

- In which state the planning starts;
- How many computational resources are assigned to it;
- Which optimisation or planning algorithm is used;
- What is the relation between the planning and the DRL algorithm.

In the other group of model-based algorithms, the model of the environment is not fully known, and the algorithms train with a learned model [70]. Normally, a representation of the environment is extracted by using supervised/unsupervised algorithms, which is carried out in a previous step as a model learning process [71–73]. Algorithms such as World Models [74] and Imagination-Augmented Agents (I2A) [75] belong to this group. Nonetheless, the accuracy of the model depends on the observable information and the capacity to adapt to changes in the model dynamics. For this reason, these algorithms are more suitable for dealing with deterministic environments. Based on the experience acquired through the interaction with the environment, three model approaches can be obtained [76,77]:

- Forward model: based on the current state and the selected action by the agent, it estimates the next state;
- Backward model: a retrospective model that predicts which state and action led to the current state;
- Inverse model: it assesses which action makes moving from one state to another.

On the other hand, model-free algorithms cannot anticipate the evolution of the environment after an action because the dynamics of the environment is unknown [78]. Thus, the algorithm estimates the most suitable action at the current state based on the acquired experience through interaction. This latter is the most frequent scenario in practice; hence, more algorithms exist [79]. The model-free DRL algorithms focus on the management of acquired experience by algorithms and how they use this information to learn a policy. This distinguishes on-policy algorithms from off-policy algorithms [80]. In the former case, the agent applies its policy generating short-term experience, which frequently consists of a fixed number of transitions (trajectory) [81,82]. Based on this information, the policy is updated, and then the experience is discarded. On the other hand, off-policy algorithms have a memory that stores the transitions created by several past policies [83]. This memory is finite and has a memory management method, for instance, FIFO (first-in, first-out) [84]. In this case, the policy is updated with a sampled batch of the stored transitions, considering the experience generated with old policies [85].

Although this latter classification is not exclusive to model-free algorithms, there is a certain parallelism with the two families of model-free algorithms represented in Figure 9, policy optimisation (PO) and Q-learning families. The first family began with the Policy Gradient algorithm and was later expanded to include the Advantage Actor Critic (A2C) [86], Asynchronous Advantage Actor Critic (A3C) [87], and proximal policy optimization (PPO) [88] algorithms. This class of algorithms is capable of handling continuous and discrete action spaces, and the action at each state is determined by a probability distribution. The second family was derived from Deep Q-Networks (DQN) [89], and algorithms such as Quantile Regression DQN (QR-DQN) [90] and hindsight experience replay (HER) [91] belong to it. In contrast to the other family, they can only deal with discrete action space environments, and the policy calculates the Q-value of each state-action pair to take a decision.

Lastly, it should be highlighted that these classifications are not exclusive, and there are algorithms that integrate features and techniques of different groups, such as the hybrid algorithms that are halfway between policy optimisation and Q-learning families (see Figure 9). Some algorithms of this group are Soft Actor-Critic (SAC) [92], Deep Deterministic Policy Gradient (DDPG) [93] and Twin Delayed Deep Deterministic Policy Gradient (TD3) [94]. These algorithms address some of the weaknesses of the other algorithms that allow the implementation of approaches to more complex problems. In addition, combinations of algorithms from different groups can be found in the literature, such as DDPG + HER [95] and model-free and model-based algorithms [61].

*Integration in the Industry*

Manufacturing involves a set of tasks that generally entail decision making by plant operators. These tasks are related to scheduling [96] (e.g., predicting the production based on future demand, guaranteeing the supply chain, planning processes to optimise production and energy consumption); process control [97] (i.e., automated processes such as assembly lines, pick-and-place and path planning); and monitoring [10] (e.g., decision support systems, calibration and quality control) [12]. As can be observed, most of these tasks are complex, and their efficient performance needs expert knowledge and time to be programmed. In the manufacturing sector, the former exists for many tasks, but the availability of time is limited even more if flexible production wants to be achieved under the framework of I4.0. Moreover, for smart factories of I5.0, other factors, such as benefits for the well-being of workers and the environment, must be considered. All in all, the automation of manufacturing tasks is a complex optimisation problem that requires novel

technologies to be addressed, such as ML. Based on a few investigations [11,98,99], below the main requirements of an ML application in the industry are listed:

- Dealing with high-dimensional problems and datasets with moderate effort;
- Capability to simplify potentially difficult outputs and establish an intuitive interaction with operators;
- Adapting to changes in the environment in a cost-effective manner, ideally with some degree of automation;
- Expanding the previous knowledge with the acquired experience;
- Ability to deal with available manufacturing data without particular needs for the initial capture of very detailed information;
- Capability to discover relevant intra- and inter-process relationships and, preferably, correlation and/or causation.

Among ML paradigms, reinforcement learning is suitable for this type of task. The trial-and-error learning through the interaction with the environment and not requiring pre-collected data and prior expert knowledge allow RL algorithms to adapt to uncertain conditions [12]. Moreover, thanks to the capacity of ANNs to create simple representations of complex inputs and functions, DRL algorithms can address complex tasks, maintaining adaptability and robustness [100]. Indeed, some applications can be found in manufacturing, for instance, in scheduling tasks [101,102] and robot manipulation [103,104].

However, the application of DRL in industrial processes presents some challenges that must be considered during the implementation. A complete list of challenges is gathered in studies such as [11,105]; however, the most common ones perceived by the authors in real-world implementations are described below.

- Stability. In industrial RL applications, the sample efficiency of off-policy algorithms is desirable. However, these show an unstable performance in high-dimensional problems, which worsens if the state and action spaces are continuous. To mitigate this deficiency, two approaches predominate: (i) reducing the brittleness to hyperparameter tuning and (ii) avoiding local optima and delayed rewards. The former can be solved by using tools that optimise the selection of hyperparameters values, such as Optuna [106], or employing algorithms that internally optimise some hyperparameters, such as SAC [107]. The other approach can be addressed by stochastic policies, for example, introducing entropy maximisation such as SAC and improved exploration strategies [108].
- Sample efficiency. Learning better policies with less experience is key for efficient RL applications in industrial processes. This is because, in many cases, the data availability is limited, and it is preferable to train an algorithm in the shortest possible time. As stated before, among model-free DRL algorithms, off-policy algorithms are more sample efficient than on-policy ones. In addition, model-based algorithms have better performance, but obtaining an accurate model of the environment is often challenging in the industry. Other alternatives to enhance sample efficiency are input remapping, which is often implemented with high-dimensional observations [109], and offline training, which consists of training the algorithm with a simulated environment [110].
- Training with real processes. Albeit training directly with the real systems is possible, it is very time consuming and entails the wear and tear of robots and automatons [105]. Moreover, human supervision is needed to guarantee safety conditions. Therefore, simulated environments are used in practice, allowing the generation of much experience at a lower cost and faster training. Nonetheless, a real gap exists between simulated and real-world environments, making applying the policy learned during the training difficult [111]. Some techniques to overcome this issue are presented in Section 5.
- Sparse reward. Manufacturing tasks usually involve a large set of steps until reaching their goal. Generally, this is modelled with a zero-reward most of the time and a high reward at the end if the goal is reached [112]. This can discourage the agent in the exploration phase, thus attaining a poor performance. To this end, some

solutions are aggregating demonstration data to the experience of the agent in a model-based RL algorithm to learn better models; including scripted policies to initialise the training, such as in QT-Opt [113] and reward shaping provides additional guidance to exploration, boosting the learning process.

- Reward function. The reward is the most important signal the agent receives because it guides the learning process [114]. For this reason, clearly specifying the goals and rewards is key to achieving a successful learning process. This becomes more complex as the task and the environment becomes more complicated, e.g., industrial environments and manufacturing tasks. To mitigate this problem, some alternatives are integrating intelligent sensors to provide more information, using heuristics techniques and replacing the reward function with a model that predicts that reward [115].

## 4. Deep Reinforcement Learning in the Production Industry

Nowadays, manufacturing industries face major challenges, such as mass customisation and shorter development cycles. Moreover, there is a need to meet the ever-rising bar for product quality and sustainability in the shortest amount of time through an ambiguous and fluctuant market demand [99]. However, those challenges also open up new opportunities for innovative technologies brought by the I4.0 and I5.0 [13,116]. Among those, AI plays a special role, and furthermore, DRL, after the outstanding results presented by OpenAI [117] and DeepMind [118], among others, is progressively shifted to the production industry [119]. In this sense, some of the main DRL features, such as the adaptability and ability to generalise and extract information from past experiences, have already been demonstrated in a few sectors, as reflected in other reviews. Among them are robotics [103,120], scheduling [121,122], cyber-physical systems [123] and energy systems [124].

Further on, in the following subsections, an overview of DRL applications in the main disciplines within the production industry is presented. These applications are usually developed in deterministic environments due to the fact that they can be modelled along with the effectiveness of DRL algorithms in them. We detail the challenges of the disciplines, frequently DRL implemented, how those are implemented and the main results. All in all, the main objective is to present the lector with different DRL solutions for the expected challenges in some of the main production industries' activities.

### 4.1. Path Planning

In manufacturing, path planning is crucial for machines such as computer numerical control (CNC) machines [125] and robot manipulation [126] to perform tasks such as painting, moving in space and welding, and additive manufacturing [110,127]. Moreover, path planning is part of the mobile robot navigation system that has an increasing presence in factories [128]. The main objective of this task is to find the optimal trajectory to move the robot or part of it from one point in space to another while maybe performing an operation. In industrial environments, other factors must be considered due to the features of the task or the environment or the potentially severe consequences of a failure. These make path planning more complex, and some of the most popular ones are the avoidance of obstacles, dynamic environments and constraints of the movements of the robots and systems.

For this application, model-free DRL algorithms are predominant, probably due to the complexity of modelling a dynamic environment [129]. Based on the analysed research articles, DQN, together with its variants, is the most used one [130–132]. Despite some issues, such as overestimating q-values and instability, DQN applications are widely used in path planning. An important task of this field is active object detection (AOD), whose purpose is to determine the optimal trajectory so that a robot has the viewpoints that allow it to gather the necessary visual information to recognise an object. DQN is still used for this purpose, outperforming other AOD methods. Fang et al. (2022) [133] recently presented a self-supervised DQN-based algorithm that improves the success rate and reduces the average trajectory length. Moreover, the developed algorithm was successfully tested with

a real robot arm. However, the applications of DQN variants need to become popular in order to overcome the aforementioned drawbacks.

Prioritised DQN (P-DQN) is used to upgrade the convergence speed of DQN, assigning more priority to those samples that contain more information in comparison with the experience [134]. These samples are more likely to be selected to update the parameters of the ANNs. Liu et al. (2022) [135] present a P-DQN-based path-planning algorithm to address path planning in very complex environments with many obstacles. This priority assignment can be detached, constituting a technique called priority experience replay (PER). This technique is combined with Double DQN (DDQN) in [136], increasing the stability of the learning process. Moreover, DDQN also offers satisfactory performance without PER. An example is the path planning application presented in [56], where the DDQN agent is pre-trained in a virtual environment with a 2D-LiDAR and then tested in a real environment using a monocular camera.

In line with I5.0, path planning has a challenge in robotic applications to achieve the estimation of time-efficient and free-collision paths. In this context, crowd navigation of mobile robots can be highlighted due to the need to predict the movement of other objects in the environment, such as humans. For this purpose, the DQN variant of Dueling DQN in combination with an online planner proposed in [137] results in equivalent or even better performance of the state-of-the-art methods (95% of success in complex environments) with less than half the computational cost. Furthermore, based on Social Spatial–Temporal Graph Convolution Network (SSTGCN), a model-based DRL algorithm is developed in [138], highlighting its robustness to changes in the environment.

Lastly, the use of hybrid DRL algorithms should be remarked on because they can work with continuous action space and are not like DQN, which is limited to discrete spaces. For example, Gao et al. (2020) [139] present a novel path planner for mobile robots that combines TD3 and the traditional path planning algorithm Probabilistic Roadmap (PRM). PRM + TD3 is trained in an incremental way, achieving an outstanding generalisation for planning long-distance paths. In addition, a variant of DDPG called mixed experience multi-agent DDPG (ME-MADDPG) is applied to coordinate the displacement of several mobile robots. This algorithm enhances the convergence properties of other DRL algorithms in this field [140].

### 4.2. Process Control

With the automation of factories, process control became a key element in manufacturing. This control is scalable from large SCADA panels that monitor the whole production chain of a factory to specific processes [141]. Moreover, this manufacturing task addresses simple control operations, such as opening valves, and complex control operations, such as coordinating several robot arms for assembling. For this purpose, control strategies have typically been applied; however, the application of artificial intelligence methods, such as neural networks, is growing thanks to the development of smart factories [10]. Given the plethora of process control tasks, this section focuses on the most recent DRL applications in this field. In addition, a subsection is dedicated to robotic control, especially robot manipulation, due to its significant role in manufacturing [142].

The literature search reflects that DRL algorithms are generally applied to control specific processes and that model-free algorithms predominate. Since control tasks usually involve continuous variables, the algorithms from the policy optimisation family and hybrid algorithms are the most used ones. Regarding the former, PPO is widely applied because it is the most cutting-edge and established algorithm within the PO family. Szarski et al. (2021) apply PPO to control the temperature in a composite curing process to reduce the cycle time [143]. The developed controller is tested with the simulation of a complex curing process in two realistic different aerospace parts, reducing up to 40% of the ramp time. Moreover, this test demonstrates the controller's applicability because it was only trained for one of the parts. Other PPO applications can be found in other manufacturing processes, such as controlling the power and velocity of a laser in charge

of melting via powder bed fusion [64] and controlling the rolls of a strip rolling process to achieve the desired flatness [144]. It should be noted that this last application is also compared with DRL hybrid algorithms, outperforming them regarding results and stability.

Although PPO has been applied to some control tasks, its on-policy nature generally entails larger training. Off-policy DRL algorithms improve it thanks to being more sample efficient [145], and DDPG is the most popular off-policy hybrid algorithm for control applications. This algorithm is an extension of DQN for continuous action spaces, and it is the first off-policy algorithm for this type of space, showing positive performance in the control of complex systems. Fusayasu et al. (2022) [146] present a novel application of DDPG in the control of multi-degree-of-freedom spherical actuators, characterised by their difficult control due to their strong non-linearities of torque. DDPG achieves a highly accurate and robust control, outperforming PID and neural network controllers. In the chemical process control, Ma et al. (2019) [147] demonstrate how a DDPG controller can control a polymerisation system, which is a complex, multi-input, non-linear chemical reaction system with a large time delay and noise tolerance. In this case, the main adaptation of the original algorithm is the inclusion of historical experience to deal with time delay. Another application of DDPG in the optimisation of chemical reactions is [148], where the maximisation of hydrogen production through the partial oxidation reaction of methane is reached. Moreover, TD3, as an improved version of DDPG, is also applied in this type of process, for instance, the multivariable control of a continuous stirred tank reactor (CSTR) [149]. The importance of DDPG and TD3 in process control in the chemical industry is shown in [150], where hybrid and PO algorithms are compared for five use cases, and DDPG and TD3 outperform all of them in all use cases.

*4.3. Robotics*

Robot manipulation encompasses a wide range of tasks, from assembly operations, such as screwing and peg-in-hole, to robot grasping and pick-and-place operations [151,152]. The characteristics of DRL make it very suitable for robotic tasks, which has produced a close relationship between both fields for many years, leading to promising results in the future [50,120]. This section includes a mini-review of the most recent DRL applications in this vast field.

Firstly, this review starts with the peg-in-hole assembly, the robotic manipulation task with the most DRL applications according to the literature search, and its high precision characterises it. For this task, PPO is the most commonly applied algorithm with applications such as [103,153,154]. Among them, the PPO controller developed by Leyendecker et al. (2021) [103] should be noted, where the algorithm is trained through curriculum learning. This technique consists of dividing the learning problem into several subtasks and learning them in ascending order of complexity, which allows the learning of the simpler tasks to be used to learn the more complex ones and improves generalisation skills [155].

Although PPO applications abound, other DRL algorithms can be found. For example, Deng et al. (2021) propose an actor-critic-based algorithm that improves the stability and sample efficiency of other state-of-the-art algorithms such as DDPG and TD3 [103,104]. In addition, training this algorithm with hierarchical reinforcement learning (HRL) notably increases the generalisation capability to other assembly tasks. HRL consists of decomposing tasks into simpler and simpler sub-tasks, establishing levels of hierarchy in which more complex parent tasks are formed by simpler child tasks. With this technique, the most basic tasks are learned, which allow for the development of more complex tasks [156]. Furthermore, among the applications of hybrid algorithms, the work of Beltran-Hernández et al. (2020) [104], which uses SAC to learn contact-rich manipulation tasks and tests the algorithm with a real robot arm, and the proposed uses of DDPG to control the force in contact-rich manipulation in [157] and to enhance the flexibility of assembly lines in [158] are noteworthy. The latter is particular in that it uses a digital twin model of the assembly line to train the DDPG algorithm, and once trained, this model is used to monitor the assembly lines and predict failures during the production stage.

Digital twins are a technology that is increasingly important in I.40 and I.5.0, which seems to be crucial to the development of smart manufacturing. Indeed, some DRL control applications, such as [158], leverages this technology to increase their data efficiency and robustness. Liu et al. (2022) train a DQN algorithm with the digital twin model of a robot arm that has to perform a grasping task [159]. In this line, Xia et al. (2021) do the same with DQN and DDQN + PER for a pick-and-place task [69]. Both cases highlighted the smoother transfer of knowledge from the simulation to the real environment thanks to digital twin models.

Finally, another robot manipulation task to which DRL is currently applied is pick-and-place, which in turn includes other tasks such as motion planning, grasping and reaching a point in space [50]. As in other robotic tasks, the use of DDPG is predominant [160]. Some recent examples are [161], whose objective is reaching a point and measuring the influence of different reward functions, and [162], where the application of DDPG results in robust grasping in pick-and-place operations. In addition, the joint use of DDPG and HER is common, highlighting the work of Marzari et al. (2021), that DDPG + HER is used together with HRL to learn complex pick-and-place tasks [163]. Nonetheless, other state-of-the-art algorithms are used in this field, such as TD3 + HER for the motion planning of robot manipulators [164] and PPO and SAC for a grasping task with an outstanding success rate [165]. In this latter work, it should be noted that SAC training requires fewer episodes, but they last longer.

### 4.4. Scheduling

The aim of scheduling is to optimise the use of time to reduce the consumption of resources in all senses, hence improving the overall efficiency of the industrial processes. In this, several sub-objectives must be considered. It plays an essential role within any kind of industry and has always been a significant research topic approached from different fields. However, due to its interdisciplinary nature, the size of the problem can easily scale up. Consequently, the optimisation problem has multiple objectives and is usually complex given the uncertainties that must be faced and the high interconnectivity of the elements involved [166]. In this sense, DRL arises as an enabling technology, as reflected in literature reviews concerning smart scheduling in the industry 4.0 framework [167].

On the one hand, in order to solve the multi-objective optimisation problem, a common approach is the implementation of multi-agent DRL algorithms. Several successful studies can be found about this in different production sectors [12]. Lin et al. (2019) [101] implemented a multi-agent DQN algorithm for a semiconductor manufacturing industry in order to cover the human-based decisions and reduce the complexity of the problem, resulting in enhanced performance. Through a similar approach, Ruiz R. et al. (2022) [102] focus on the maintenance scheduling of several machines presenting up to $\approx 75\%$ improvement in overall performance. Other studies combine those algorithms with IoT devices for smart resource allocation [168] or with other algorithms, such as Lamarckian local search for emergency scheduling activities [169]. For the latter, Baer et al. (2019) [170] propose an interesting approach by implementing a multi-stage learning strategy, training different agents individually but optimising them together towards the global goal, presenting great results. On the other hand, in order to face the increasing fluctuation in production demand and product customisation, actor-critic DRL approaches are usually implemented [171].

The actor-critic approach is characterised by its robustness [172] and acts as an upgrade of the traditional Q-learning, which could act as a decision-support system easing operators scheduling tasks [173,174]. Through the actor-critic approach, the policy is periodically checked and recalibrated to the situation, which highly increases the adaptability and eases the implementation in real-time scheduling [96,175]. In addition, several studies reflect that it can be implemented with cloud-fog computing services [176,177]. Furthermore, the performance can be increased by implementing a processing approach divided into batches, as reflected in Palombarini et al. (2018, 2019) studies [178,179]. There are also some novel approaches integrating different neuronal networks that aim to cope with complexity and

expand the applications. For example, Park et al. (2020) implemented a proximal policy optimisation (PPO) neuronal network trained with relevant information from scheduled processes, such as the setup status [180].

For latter, despite the great results presented by the research, unfortunately, most of those approaches are not adopted in a practical context. Due to the scheduling policies already established in the production industries, it is quite complex to introduce novel approaches even if the research shows good results. Consequently, increasing research efforts are required in this direction.

### 4.5. Maintenance

The maintenance objective is to reduce breakdowns and promote overall reliability and efficiency [181]. The term mainly refers to tasks required to restore full operability, such as repairing or replacing damaged components. It significantly impacts the operational reliability and service life of the machinery in any industry. There are four types of maintenance: reactive, preventive, predictive and reliability-centred [182,183]. Historically, reactive maintenance has predominated, which was performed after the failure of the machine, mainly due to limited knowledge about their operation and failures. Nowadays, this strategy is still in use for unpredictable failures and failures of cheap objects. Over time, the understanding of the process has increased, and preventive maintenance has come up. Further on, I4.0 technologies and advances in AI have enabled predictive and reliability-centred maintenance [184].

As part of AI advances for maintenance activities in the industry, RL algorithms play an important role due to their self-learning capability [185]. Moreover, the integration of neuronal networks, resulting in DRL, expands the applications and performances even further [186]. Their application can help anticipate failures by predicting key parameters and also prevent failures through in-line maintenance, enlarging the lifetime of components.

The anticipation of failures is usually combined with scheduling optimisation to maximise the results [187,188]. In order to speed up the learning phase, Ong, K.S.H, et al. (2022) boards the predictive maintenance problem with a model-free DRL conjoined with the transfer learning method to assist the learning by incorporating expert demonstrations, reducing the training phase time by 58% compared with baseline methods [189]. On the other hand, Acernese, A. et al. board fault detection for a steel plant through a double deep-Q network (DDQN) with prioritised experience replay to enhance and speed up the training [190].

There are also hybrid approaches, such as the one proposed by Chen Li et al. (2022) [191], where feedback control is implemented based on an advantage actor-critic (A2C) RL algorithm to predict the machine status and control the cycle time accordingly. In addition, Yousefi, N. et al. (2022), in their study, propose a dynamic maintenance model based on a Deep Q-learning algorithm to find the optimal maintenance policy at each degradation level of the machine's components [192].

### 4.6. Energy Management

Nowadays, and especially with the I5.0 and worldwide policies (e.g., Paris agreement [193]), energy consumption and environmental impact are in the spotlight. In this sense, AI algorithms such as DRL can boost energy efficiency and reduce the environmental impact of the manufacturing industry [194]. The algorithms are usually implemented into the energy market to reduce costs and energy flow control in storage and machines operation to increase their energy consumption effectiveness [195]. In resource- and energy-intensive industries such as printed circuit boards (PCB) fabrication, Leng et al. demonstrated that the DRL algorithm was able to improve lead time and cost while increasing revenues and reducing carbon use when compared to traditional methods (FIFO, random forest) [196]. Lu R. et al. (2020) faced a multi-agent DRL algorithm against a conventional mathematical modelling method simulating the manufacturing of a lithium–ion battery. The benchmark presents a 10% reduction in energy consumption [197].

## 5. Simulation-to-Reality Transfer

Deep reinforcement learning is a relatively novel technology with a promising future in the industry field. However, its major challenge is the implementation in real-world tasks, which demands an increase in the stability, sample efficiency and generalisation of DRL algorithms [198]. To improve these aspects and avoid the drawbacks of training with real environments, the current trend is to first train the algorithm in simulated environments and then test it in real environments [199]. Nonetheless, the knowledge transfer from simulation to reality is not straightforward. To this end, there is an area in reinforcement learning called sim-to-real transfer that encompasses a whole ream of techniques to achieve an effective learning transfer [200]. This section presents the most used ones in industrial DRL applications.

The selection of the simulator is crucial for attaining a successful transfer from simulation to reality [200]. The more realistic a simulation is, the better performance of the algorithm may be expected in real-world tasks. Additionally, in industrial use cases, many factors must be considered due to their complexity, such as physic simulations, virtual representations of objects, recreations of sensor data and artificial lightning [201]. Gazebo [202], Unity3D [203], PyBullet [204] and MuJoCo [205] are the most popular simulators in the literature thanks to their accurate physics engine and customisable environments. These platforms allow for creating customised environments, loading pre-existing models of robots and systems and simulating the interaction between them and other elements. In addition, benchmark suites can be loaded into those platforms, such as Arena-bench [206], which allows training, testing and evaluating navigation algorithms for dynamic obstacle avoidance. Regarding the performance of the aforementioned simulators, Gazebo and Unity3D offer highly realistic simulations of complex scenarios, while PyBullet and MuJoCo slightly reduce these features in exchange for faster simulations. Table 1 gathers some examples of DRL applications trained to learn industrial tasks in those simulators. Lastly, although they are not simulators as such, special mention should be made of digital twin models for training, as they improve the performance of the algorithm in real-world tasks by improving the simulation environment [159].

**Table 1.** DRL applications trained in different simulators.

| Simulator | Example |
|---|---|
| Gazebo | • Robotic Grasping using Deep Reinforcement Learning [207]<br>• End-To-End Autonomous Exploration for Mobile Robots in Unknown Environments through Deep Reinforcement Learning [208]<br>• An Efficient Deep Reinforcement Learning Framework for UAVs [209]<br>• Path Planning of Mobile Robot Using Reinforcement Learning [210] |
| Unity3D | • Goal-Oriented Obstacle Avoidance by Two-Wheeled Self Balancing Robot [211]<br>• KIcker: An Industrial Drive and Control Foosball System automated with Deep Reinforcement Learning [212]<br>• Research on Autonomous Navigation Control of Unmanned Ship Based on Unity3D [213]<br>• Crowd Navigation in an Unknown and Dynamic Environment Based on Deep Reinforcement Learning [214]<br>• Research on robot arm control based on Unity3D machine learning [215] |
| PyBullet | • Deep Reinforcement Learning Based Trajectory Planning Under Uncertain Constraints [216]<br>• Robotic Lever Manipulation using Hindsight Experience Replay and Shapley Additive Explanations [217]<br>• Robust Quadruped Jumping via Deep Reinforcement Learning [218] |
| MuJoCo | • Learning Continuous Control Actions for Robotic Grasping with Reinforcement Learning [219]<br>• MANGA: Method Agnostic Neural-policy Generalization and Adaptation [220]<br>• Diversity-Driven Exploration Strategy for Deep Reinforcement Learning [221] |

In most cases, training with a realistic simulated environment and an appropriate hyperparameter setting of the algorithm does not guarantee the successful performance of the algorithm in the real task. For this purpose, there are numerous sim-to-real techniques that help to bridge the gap between simulation and reality. Based on the literature search, three

techniques have been found to be used in DRL applications in manufacturing, of which domain randomisation is the most extended. These techniques are briefly described below, along with a list of applications in which they are used both solely and in combination (see Table 2).

- System identification. This is a wide field that involves creating a precise mathematical representation of the actual world in order to increase the realism of the simulation [222]. To this end, the observed data (inputs and outputs) and the knowledge of the process dynamics are used to build the model [223]. This approach has certain drawbacks, which are exacerbated when applied to industrial processes, such as the need for parameter calibration, data gathering and modelling of how external influences affect the operation of the robot (e.g., the wear-and-tear of its joints) [224]. Nonetheless, as observed in the aforementioned applications [69,158,159], the development of digital twin models is trending in the building of accurate mathematical models, becoming a fundamental pillar in smart manufacturing [225].

- Domain randomisation. To accommodate a wider variety of environmental setups and potential scenarios, simulation parameters are randomly generated [226]. This includes two groups of techniques that bridge the gap between the actual world and the virtual one. On the one hand, visual randomisation addresses the inclusion of randomness in the observation, such as changing the camera location, artificial light, and textures [227]. On the other hand, dynamics randomisation entails changing the simulator's physical settings, such as item sizes, friction factors, or joint motor characteristics [228]. Further, the variation of these parameters is not simply random, and its distribution affects the performance of the algorithm [229]. Indeed, many variants are proposed to obtain more robust performance, such as Active Domain Randomisation [230] and Neural Posterior Domain Randomisation (NPDR) [231].

- Domain adaptation. It aims to harmonise the spaces in order to bridge the gap between the simulated and actual environments. Although action, reward and transition spaces are often similar in simulation and reality, state spaces show more pronounced differences [232]. This is mainly due to the tools for perception and feature extraction employed [233]. For this reason, this technique can be found in many applications as state adaptation. Among the most popular domain adaptation methods, the discrepancy-based [234], adversarial-based [235] and reconstruction-based approaches [236] are noteworthy.

**Table 2.** Sim-to-real transfer methods used in DRL applications in manufacturing.

| | Domain Randomisation | Domain Adaptation | System Identification |
|---|---|---|---|
| Deep Reinforcement Learning for Robotic Control in High-Dexterity Assembly Tasks—A Reward Curriculum Approach [103] | ✓ | | |
| Towards Real-World Force-Sensitive Robotic Assembly through Deep Reinforcement Learning in Simulations [154] | ✓ | | |
| Variable Compliance Control for Robotic Peg-in-Hole Assembly: A Deep-Reinforcement-Learning Approach [104] | ✓ | | |
| A flexible manufacturing assembly system with deep reinforcement learning [158] | | | ✓ |
| A digital twin-based sim-to-real transfer for deep reinforcement learning-enabled industrial robot grasping [159] | | | ✓ |
| A digital twin to train deep reinforcement learning agent for smart manufacturing plants: Environment, interfaces and intelligence [69] | | | ✓ |
| Learning to Centralise Dual-Arm Assembly [237] | ✓ | | |

**Table 2.** *Cont.*

| | Domain Randomisation | Domain Adaptation | System Identification |
|---|:---:|:---:|:---:|
| Sim-to-Real Visual Grasping via State Representation Learning Based on Combining Pixel-Level and Feature-Level Domain Adaptation [238] | | ✓ | |
| Reinforcement Learning Experiments and Benchmark for Solving Robotic Reaching Tasks [110] | ✓ | | |
| Preparing for the Unknown: Learning a Universal Policy with Online System Identification [239] | | | ✓ |
| MANGA: Method Agnostic Neural-policy Generalisation and Adaptation [220] | ✓ | | |
| Sim-to-real transfer reinforcement learning for control of thermal effects of an atmospheric pressure plasma jet [240] | ✓ | | |
| Policy Transfer via Kinematic Domain Randomisation and Adaptation [241] | ✓ | ✓ | |
| Latent Attention Augmentation for Robust Autonomous Driving Policies [242] | | ✓ | |

## 6. Conclusions and Future Work

Within this review, the challenges the manufacturing industry faces nowadays have first been introduced. However, those challenges also open up new opportunities for the innovative technologies raised with the I4.0 and I5.0. Among those, the AI and particularly the DRL algorithms have been remarked on, an ideal solution for the unpredictable and fluctuant changes in the current demand. Through the introduction of the RL concepts and expansion of those with the DNNs towards DRL, the potential and variability of those kinds of algorithms have been highlighted. Furthermore, since those algorithms are data based, they can be easily reconfigured to adapt to the industry processes' needs. The main requirement is access to data from the environment, which, nowadays, with the IoT devices and monitoring systems, is not a problem in most of the manufacturing industries. Moreover, the implementation of new concepts, such as the digital twins, in response to a missing model of the environment, will boost the performance and application of DRL algorithms even further.

The application of DRL algorithms is found all across the manufacturing industry activities. In one of the major fields, robotics, the review reflects that the performance has improved significantly by implementing DRL and shows tremendous promise for completing challenging robotic manipulation tasks. Nevertheless, because of the potentially dangerous but insufficient interactions, a significant amount of work is still required before DRL algorithms can be used directly in real-world jobs. Analysis of the sample efficiency, stability and generalisability of RL algorithms is also still lacking. Sim-to-real has emerged as a promising option as a result of its analysis of RL algorithms in simulation and subsequent implementation in real-world activities.

Nevertheless, there are still a few challenges to face in order to fully exploit the potential of the DRL algorithms. The first challenge is the selection of algorithm. As reflected in the review, there are many different algorithms, and selecting the most suitable one for the problem is not easy. In this sense, this review aims to ease the selection of the most suitable algorithms by presenting to the lector several examples of problems within a manufacturing industry and how they have been solved by implementing DRL approaches. The second challenge is the implementation in the production. Unfortunately, the results presented in most of the investigations come from simulations, and they are not transferred to real-world scenarios due to industries' working policies. Therefore, despite the great results presented, it is important to bear in mind that the performance would probably reduce when transferring to a real-world scenario. Moreover, an update of the industrial

digital infrastructure is also required in order to ease the integration of novel digital tools such as the DRL algorithms. Concerning this challenge, the review presented in Section 5 in regard to sim-to-real transfer tools could help the reader-developers go one step further in their validation process and boost the transfer to a real-world scenario.

Overall, to face the existing challenges and fully exploit DRL capacity, the next steps have been proposed. In the first instance, and this is the main point, validations need to be close to real environments: enhance the simulation of the real environment to ensure the fulfilment of the problem's specifications, improve the training phase and accelerate the deployment. In this sense, tools such as digital twins could help. Moreover, DRL algorithms have to include an evaluation of policy safety and robustness. In the second instance, standardise the implementation process. Defining common and easily understandable guides on how to implement and test DRL algorithms in a real environment will boost the validation and help the industries understand the DRL algorithms easing their deployment in testing in their facilities. In conclusion, the next steps should focus on accelerating and facilitating the validation of the algorithms in real environments to boost their deployment.

In a nutshell, the review spotlights that DRL applicability is observed across all activities within the manufacturing industries, outperforming the conventional techniques and, most importantly, boosting the resilience and adaptability of the manufacturing process. However, there is still much work to be carried out, in both academics and industries, to fully exploit the potential of those disruptive tools, start the deployment in the industries, and take a further step towards the I5.0 transformation of industries.

**Author Contributions:** Conceptualisation, A.d.R.T.; investigation, methodology and writing—original draft preparation, A.d.R.T., D.S.A. and Á.O.R.; software, visualisation and writing—original draft preparation, A.H.B.; writing—review and editing and supervision, A.d.R.T. and L.E.A.G.; All authors have read and agreed to the published version of the manuscript.

**Funding:** This research received no external funding.

**Institutional Review Board Statement:** Not applicable.

**Informed Consent Statement:** Not applicable.

**Acknowledgments:** This publication has emanated as part of the research and activities done along MORSE project which has received funding from the European Union's Horizon 2020 research and innovation program under grant agreement No. 768652.

**Conflicts of Interest:** The authors declare no conflict of interest.

## Abbreviations

| | |
|---|---|
| A2C | Advantage Actor-Critic |
| A3C | Asynchronous Advantage Actor-Critic |
| AI | Artificial intelligence |
| ANN | Artificial neural network |
| AOD | Active object detection |
| CNC | Computer numerical control |
| CPS | Cyber-physical Systems |
| CSTR | Continuous stirred tank reactor |
| DDPG | Deep Deterministic Policy Gradient |
| DDQN | Double DQN |
| DNN | Deep neural network |
| DQN | Deep Q-Networks |
| DRL | Deep reinforcement learning |
| FIFO | First-in first-out |
| GDP | Gross domestic product |
| HER | Hindsight experience replay |
| HRL | Hierarchical reinforcement learning |
| I2A | Imagination-Augmented Agents |

| I4.0 | Industry 4.0 |
|------|--------------|
| I5.0 | Industry 5.0 |
| IoT | Internet of Things |
| MDP | Markov Decision Process |
| ME-MADDPG | Mixed Experience Multi-agent DDPG |
| ML | Machine learning |
| NPDR | Neural Posterior Domain Randomisation |
| PCB | Printed circuit board |
| P-DQN | Prioritised DQN |
| PER | Priority experience replay |
| PID | Proportional-integral-derivative |
| PO | Policy optimisation |
| PPO | Proximal policy optimisation |
| PRM | Probabilistic Roadmap |
| QR-DQN | Quantile Regression DQN |
| RL | Reinforcement learning |
| SAC | Soft Actor-Critic |
| SSTGCN | Social Spatial–Temporal Graph Convolution Network |
| TD3 | Twin Delayed Deep Deterministic Policy Gradient |

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
