# Peer review of "A Review of Deep Reinforcement Learning Approaches for Smart Manufacturing in Industry 4.0 and 5.0 Framework"

_applsci, doi:10.3390/app122312377_

Round 1

Reviewer 1 Report

This paper presents a review of the application of reinforcement learning in industry 4.0 and 5.0, here are some comments:

·        It is recommended to point out the main contributions of the review, as well as why a new review in the medium is important. This will help differentiate it from other papers.

·        In the presented framework of reinforcement learning, it is recommended to mention the formal definition of the Markov process.

·        In the framework, there is no mention of stochastic state transitions, the study is limited to deterministic transitions, which can reduce the scope of RL applications in the industry.

·        minor revision in English recommended

·        in line 187 you should refer to figure 3 instead of figure 2

·        It is recommended to include in the conclusions issues related to multi-agent systems that apply reinforcement learning

Author Response

Thank you very much for your review.

  • It is recommended to point out the main contributions of the review, as well as why a new review in the medium is important. This will help differentiate it from other papers.

Done. There has been added more context about this in the introduction section.

  • In the presented framework of reinforcement learning, it is recommended to mention the formal definition of the Markov process.

It has been mentioned.

  • In the framework, there is no mention of stochastic state transitions, the study is limited to deterministic transitions, which can reduce the scope of RL applications in the industry.

The application studied are usually developed in a deterministic environment given the nature of the problems within the industry. However, there has been added some more details concerning this matter. Please check the attached updated draft.

  • minor revision in English recommended

Done. Please check the attached updated draft.

  • in line 187 you should refer to figure 3 instead of figure 2

It has been corrected in the new draft. 

  • It is recommended to include in the conclusions issues related to multi-agent systems that apply reinforcement learning

Please check the attached updated draft.

Reviewer 2 Report

Applied Sciences (ISSN 2076-3417)

applsci-2069443

A review of Deep Reinforcement Learning approaches for smart manufacturing in Industry 4.0 and 5.0 framework.

Dear Authors,

Good work (paper). Congratulations to the authors. A very topical subject.

However, I have some comments:

§  In References there is a lot of variation in the way they are cited, some names are capitalized (all BIG letters)

§  There are a lot of abbreviations in this paper, please explain them at the end: add a glossary of abbreviations alphabetically to the paper.

§  Figure 8. Grid not visible, could make two separate drawings a) grid, b) Venn

§  Figure 5. Source clustering through Bradford's Law.    not visible, not visible, The font is too small

§  Figure 3. Three-field plot:                 My ask: Can the font be black

§  In my opinion, according to topic of paper: A review of Deep Reinforcement Learning approaches for 2 smart manufacturing in Industry 4.0 and 5.0 framework,   the authors write little about Industry 5.0. I propose add more references about I 5.0, and more information in the text. 

§  Other authors also performed bibliometric analyses of I 4.and Systematic Review of I 5.0, worth adding to the paper.

§  In my opinion approaches for smart manufacturing in Industry 4.0 and 5.0 framework are different because in I 4.0 the key  changes are the technology pillars and in I 5.0 the usability of technology for the environment and people.

§  In my opinion the topic of the paper suggests that authors are more likely to accept differences or the evolution of smart manufacturing.

These comments are not large and can improve the paper.

Best wishes

Reviewer

Author Response

Thank you very much for your review.

  • In References there is a lot of variation in the way they are cited, some names are capitalized (all BIG letters)

It has been corrected. 

  • There are a lot of abbreviations in this paper, please explain them at the end: add a glossary of abbreviations alphabetically to the paper.

There has been added a glossary table as suggested. 

  • Figure 8. Grid not visible, could make two separate drawings a) grid, b) Venn

Done.

  • Figure 5. Source clustering through Bradford's Law.    not visible, not visible, The font is too small

Corrected. Now should be more visible. 

  • Figure 3. Three-field plot:                 My ask: Can the font be black

We have tried to make the text more visible, however, there is no possibility to change the font to black. 

  • In my opinion, according to topic of paper: A review of Deep Reinforcement Learning approaches for 2 smart manufacturing in Industry 4.0 and 5.0 framework,   the authors write little about Industry 5.0. I propose add more references about I 5.0, and more information in the text. 

There have been added more references and text concerning this. 

  • Other authors also performed bibliometric analyses of I 4.and Systematic Review of I 5.0, worth adding to the paper.

Please check the attached document (updated draft). 

  • In my opinion approaches for smart manufacturing in Industry 4.0 and 5.0 framework are different because in I 4.0 the key  changes are the technology pillars and in I 5.0 the usability of technology for the environment and people.

Still, there are key enabling technologies, such as AI (and more concretely DRL), which are promoted within both frameworks. Even though the overall objective may differ. 

  • In my opinion the topic of the paper suggests that authors are more likely to accept differences or the evolution of smart manufacturing.

Please check the attached document (updated draft).